# The Health and Healthcare Outcomes of Trans and/or Non-Binary Adults in England: Protocol for an Analysis of Responses to the 2021 GP Patient Survey

Catherine L. Saunders [1,2,*], Jenny Lund [1], Amy M. Mason [3], Meg Roberts [4], Jack Smith [5] and Robbie Duschinsky [1]

1 Primary Care Unit, Department of Public Health and Primary Care, University of Cambridge, Cambridge CB2 0SR, UK; jl897@medschl.cam.ac.uk (J.L.); rd522@medschl.cam.ac.uk (R.D.)
2 Cambridge Public Health, University of Cambridge, Cambridge CB2 0SR, UK
3 British Heart Foundation Cardiovascular Epidemiology Unit, Department of Public Health and Primary Care, University of Cambridge, Cambridge CB1 8RN, UK; am2609@medschl.cam.ac.uk
4 Newnham College, University of Cambridge, Cambridge CB3 9DF, UK; mr730@cam.ac.uk
5 LifeStrong, Wolverhampton WV4 5UH, UK; j-a-c-k-smith@hotmail.co.uk
* Correspondence: ks659@medschl.cam.ac.uk

**Abstract:** Background: The large-scale quantitative evidence base to understand and improve health and healthcare outcomes for people who are trans and/or non-binary is still developing, although what research there is suggests that risk of poor health is high, and experiences of healthcare services are often poor. In 2021 the GP Patient Survey, which is carried out annually to measure patient experience in primary care in England, added inclusive questions about gender identity and trans status for the first time. Methods: This protocol paper pre-registers the methods that we will use for this work for a secondary analysis of these data, including both the statistical analysis protocol and early patient and public involvement work, to answer the following three research questions: (1) What are the (a) demographic characteristics, (b) health conditions, and (c) healthcare experiences of trans and/or non-binary adults in England? (2) Was there any difference in whether people who are trans and/or non-binary had been asked to shield during the COVID-19 pandemic or not compared with all other survey responders? (3) Does the relationship between being trans and/or non-binary, and self-reported long-term mental health problems, autism and autistic spectrum disorder and learning disability vary by age, gender, ethnicity, deprivation, sexual orientation or region?

**Keywords:** trans; transgender; patient experience; primary care; healthcare outcomes; health

## 1. Background

The large-scale quantitative evidence base to understand the health and healthcare outcomes for people who are trans and/or non-binary is still developing [1], although what research there is suggests that risk of poor health is high, and experiences of healthcare services are often poor [2–7].

During 2020 we carried out a collaborative health research prioritisation exercise, working with an LGBTQ+ (LGBTQ+ here stands for Lesbian, Gay, Bisexual, Transgender, Queer and other identities) patient and public involvement (PPI) panel to identify the most important areas for future health research. The top three research themes prioritised by the panel were health service delivery (particularly primary care), the prevention of ill-health (including mental health), and research understanding the intersectionality of sexual orientation and gender identity with other disadvantaged identities [8]. Within these three themes, from a policy perspective, mental health research remains a particular priority; the Academy of Medical Sciences Mental Health Research Goals 2020–2030 included a specific target to understand the barriers to health service access across different

communities, including for people who are LGBTQ+, and people living with intersectional disadvantage [9]. The UK Health Research Classification System includes autistic spectrum disorders and learning disability within this mental health research grouping [10].

The evidence needed to address these research priorities has been limited in part by poor recording of gender identity and trans status in routine data sources used for research. Risk prediction tools, which are used in primary care practice to target interventions to prevent ill health among people who are at high risk, are an area where lack of data has had a particular impact on developing appropriate approaches for trans and/or non-binary adults [11]. Of recent relevance, during the COVID-19 pandemic the QCovid risk prediction tool used primary care data to predict whether someone was likely to be at high risk of a poor outcome from COVID-19 infection [12]. People who were identified were sent letters advising them to "shield" at home and avoid social contact to minimise risk. One of the stated limitations of the QCovid model is that it is not appropriate for use for people who are trans; the guidance for clinicians states that the research to develop the algorithm was carried out only using sex registered at birth [13]. This could have the consequence of trans populations either being asked inappropriately to shield, or not to have been invited to shield even when this would have been the most appropriate course of action. Evidence on the experience of LGBTQ+ communities during the pandemic to date has mainly been based on data from online convenience samples [14].

However, in 2021 the GP Patient Survey, which is carried out annually to measure patient experience in primary care in England, added inclusive questions about gender identity and trans status for the first time. The GP Patient Survey provides nationally representative estimates for all adults age 16+ registered with a GP in England (registration is almost universal at over 97%).

In the research described in this protocol we propose a secondary data analysis of the 2021 GP Patient Survey to provide evidence to address these three collaboratively developed research priorities. This research aims to provide evidence which will be relevant for GPs and other professionals planning healthcare services and working with trans and/or non-binary adults in community settings, building on previous work including that of Ellis et al., which considered mental health and gender identity clinical settings [4].

Although there are a small number of gender identity specific healthcare services in England, most healthcare contacts are in primary care; and most healthcare contacts for people who are trans and/or non-binary will not be specifically about gender identity. Research is needed to understand how best to understand the primary healthcare needs and support people who are trans and/or non-binary in primary care practice [15]. There are specific healthcare issues for people who are trans or non-binary, which are managed in primary care including ensuring appropriate access to certain screening programmes, such as cervical cancer or abdominal aneurysm screening [16], or the possible continuation of hormone prescribing initiated in specialist services. Our analyses here focus on population-based estimates health and healthcare outcomes seen by or managed in primary care rather than a specific focus on transgender health services.

We will answer three research questions:

1. What are the (a) demographic characteristics, (b) health conditions, and (c) healthcare experiences of trans and/or non-binary adults in England?
2. Was there any difference in whether people who are trans and/or non-binary had been asked to shield or not compared with all other survey responders?
3. Does the relationship between being trans and/or non-binary, and self-reported long-term mental health problems, autism and autistic spectrum disorder and learning disability vary by age, gender, ethnicity, deprivation, sexual orientation or region?

## 2. Methods
*Involvement of Public Contributors/Experts by Experience*

The research ideas for this analysis have been shaped by earlier collaborative research prioritisation work with an LGBTQ+ public involvement panel, and for research question 3

the specifications of the research funding call ("Improving mental health and wellbeing in underserved populations through collaborative research", NIHR three Schools). The proposed analyses therefore contain research ideas relating to research question 3 which have not been co-produced, and although we have addressed this in part through collaboration with the panel at this protocol development stage, we acknowledge as a limitation.

At this protocol development stage the research team met with four members of this LGBTQ+ panel who had an interest in this specific research topic, including two trans panel members and two who identify as queer. Following these initial conversations, a fifth non-binary member will join the panel for future meetings. Panel members are paid in accordance with NIHR Involve guidelines, and the research team is being advised by a professional PPI facilitator to ensure best practice in this public involvement work.

The panel welcomed the research and its goals, but also noted the strong limitation that neither the lead researcher (CS) nor any of the academic research team are trans. This included both the practical "not another research project about people who are trans by someone who isn't" and the more central concern that the emotional work of understanding trans experience as part of the research is being held by the lay panel rather than academic team. One panel member (although supportive of the research as a whole) decided not to continue with the project as a result of this concern.

We acknowledge this as a limitation to this work (as it may also be for research in other areas as well), but it is not one that it is possible to address within the scope of the available funding for this project without asking for unremunerated labour by a trans or non-binary academic.

At the suggestion of one panel member, the team contacted Tash Oakes-Monger at NHS England, who will support this work, particularly around ensuring that thoughtful and correct language is used in all reporting.

The panel also highlighted the importance of care with the analyses looking in particular at autism, and to be aware about the potentially transphobic media and societal context that this work will appear in. As some mitigation for this concern, it is important to reiterate that the over-arching aim of this work is to provide evidence to support the improvement of primary care practice and delivery, and all work will be developed within this contextual frame.

Further areas where the PPI panel input influenced this protocol are described below in the methods section in the relevant place.

### 3. Data

In January 2021 the GP Patient Survey was sent by post to 2,408,303 adult patients registered with a GP in England, from 6694 GP practices, followed by an SMS reminder and two further postal mailings to initial non-responders. Patients from GP Practices with historically low response rates are oversampled. Paper and online responses were possible (36% of responses were online in 2021). Responses are weighted to account for design, non-response and calibration to the population of eligible patients. Full details are in the study technical guidance [17].

### 4. Survey Measures

#### 4.1. Measuring Gender Identity

In 2021, for the first time, the GP Patient Survey included revised questions covering sex, gender identity and gender reassignment in order for NHS England to meet its duties under the Equality Act (2010) to collect data and address health inequalities in relation to both sex and gender reassignment. It is a strength that survey data collections are able to make this change, as updates to clinical systems are slower, and primary and secondary care data systems are generally not yet able to comprehensively collect both gender and sex [18]. To our knowledge this is the first study to use these responses.

The new questions were developed in consultation with NHS England and stakeholders and were tested in interviews with patients, including trans and non-binary patients,

during September 2020. In addition, the questionnaire was reviewed by the Plain English Campaign to meet Plain English criteria; a set of principles designed to ensure information is presented clearly. The questions included are similar to those asked in the 2021 English Census, which underwent a large amount of methodological development work, but additionally ask about non-binary identities and additionally allow respondents to self-describe their gender.

The two included questions ask:

(1) "Which of the following best describes you?" With response options "Female", "Male", "Non-binary", "Prefer to self-describe", and "Prefer not to say".

(2) "Is your gender identity the same as the sex you were registered at birth?" with response options "Yes", "No", and "Prefer not to say".

Paper survey respondents would have been able to endorse more than one option for each question, however where they did these were recoded as missing, while online responses were restricted to one option per question.

### 4.2. Measuring Long-Term Physical and Mental Health Conditions

Respondents were asked "Which, if any, of the following long-term conditions do you have?" Prevalence estimates for long-term health conditions based on responses to a slightly earlier version of this question have been found comparable with nationally representative data sources from England [19] and responses for "a long-term mental health problem" option are used in national public health data collections [20].

### 4.3. Measuring Patient Experience

The GP Patient Survey questions were developed originally by the University of Cambridge and the University of Exeter [21,22], although they are regularly reviewed and updated. Questions ask about experiences of primary care across four main domains of healthcare quality: access, continuity, communication and overall experiences of care, with Likert scale response options. The full question wording for patient experience items are available in the questionnaire [23] and items to be included in the analysis are summarised (along with the categorisation of the response options for the primary analysis) in Table 1.

**Table 1.** Patient experience, question wording.

| **Overall experience** | |
| --- | --- |
| Overall experience | *Overall, how would you describe your experience of your GP practice?* Very good, Fairly good, compared with Neither good nor poor, Fairly poor, and Very poor |
| Overall experience of making an appointment | *Overall, how would you describe your experience of making an appointment?* Very good, Fairly good, compared with Neither good nor poor, Fairly poor, and Very poor |
| **Before trying to make an appointment** | |
| Tried self-management | *Before you tried to get this appointment, did you do any of the following?* Spoke to a pharmacist, Tried to treat myself/the person I was making this appointment for (for example with medication), compared with all other question respondents * |
| Asked friends or family | *Before you tried to get this appointment, did you do any of the following?* Asked for advice from a friend or family member, compared with all other question respondents * |
| Tried online, telephone or other NHS services | *Before you tried to get this appointment, did you do any of the following?* Used an online NHS service (including NHS 111 online), Called an NHS helpline, such as NHS 111, Contacted or used another NHS service, compared with all other question respondents * |
| Tried online or other non-NHS services | *Before you tried to get this appointment, did you do any of the following?* Used a non-NHS online service, or looked online for information, Tried to get information or advice elsewhere (from a non-NHS service), compared with all other question respondents * |
| **Access** | |
| Found GP practice website easy to use | *How easy is it to use your GP practice's website to look for information or access services?* Very easy, Fairly easy, compared with Not very easy, and not at all easy. ** |

**Table 1.** *Cont.*

| | |
|---|---|
| Tried to make an appointment in the last 6 months | *When did you last try to make a general practice appointment, either for yourself or for someone else?* In the past 3 months, Between 3 and 6 months ago, compared with Between 6 and 12 months ago, More than 12 months ago, and I haven't tried to make an appointment since being registered with my current GP practice. |
| Getting through on the phone | *Generally, how easy is it to get through to someone at your GP practice on the phone?* Very easy, Fairly easy, compared with Not very easy, Not at all easy |
| Helpful receptionists | *How helpful do you find the receptionists at your GP practice?* Very helpful, Fairly helpful, compared with Not very helpful, and Not at all helpful |
| Offered a choice when booking appointment | *On this occasion, were you offered any of the following choices of appointment?* Yes, a choice of place (for an appointment in person), Yes, a choice of time or day, Yes, a choice of healthcare professional, Yes a choice of type of appointment (phone call, online, video call, in person), compared with None of these |
| Satisfied with appointment times available | *How satisfied are you with the general practice appointment times that are available to you?* Very satisfied, Fairly satisfied, compared with Neither satisfied nor dissatisfied, Fairly dissatisfied, and Very dissatisfied |
| Offered an acceptable appointment | *Were you satisfied with the appointment (or appointments) you were offered?* Yes and I accepted an appointment, No, but I still took an appointment, compared with No, and I did not take an appointment, and I was not offered an appointment |
| Satisfied with appointment offered | *Were you satisfied with the appointment (or appointments) you were offered?* Yes and I accepted an appointment, compared with No, but I still took an appointment, No, and I did not take an appointment, and I was not offered an appointment |
| Remote appointment (telephone or online) | *What type of appointment did you get?* I got an appointment . . . " . . . to speak to someone on the phone", " . . . to speak to someone online (for example on a video call)" compared with " . . . to see someone at my GP practice", " . . . to see someone at another general practice location", " . . . for a home visit" |
| **Continuity** | |
| Have a preferred GP | *Is there a particular GP you usually prefer to see or speak to?* Yes, for all appointments, Yes for some appointments but not others, compared with No *** |
| Able to see preferred GP | *How often do you see or speak to your preferred GP when you would like to?* Always or almost always, A lot of the time, compared with Some of the time, and Never or almost never. **** |
| **Communication** | |
| Involved in decisions about care and treatment | *During your last general practice appointment, were you involved as much as you wanted to be in decisions about your care and treatment?* Yes, definitely, Yes, to some extent, compared with No, not at all |
| Had MH needs in last appointment | *During your last general practice appointment, did you feel that the healthcare professional recognised and/or understood any mental health needs that you might have had?* Yes, definitely, Yes to some extent, No, not at all, compared with I did not have any mental health needs |
| MH needs recognised and understood | *During your last general practice appointment, did you feel that the healthcare professional recognised and/or understood any mental health needs that you might have had?* Yes, definitely, Yes to some extent, compared with No, not at all |
| Confidence and trust | *During your last general practice appointment, did you have confidence and trust in the healthcare professional you saw or spoke to?* Yes, definitely, Yes, to some extent, compared with No, not at all |
| Needs were met | *Thinking about the reason for your last general practice appointment, were your needs met?* Yes, definitely, Yes, to some extent, compared with No, not at all |

\* Only including respondents who have tried to make an appointment with their GP. \*\* Only including respondents who have tried to use their GP practice's website. \*\*\* Excluding responses from people with only one GP in their practice. \*\*\*\* Only including people with a preference for a particular GP.

*4.4. Shielding*

During 2020, the QCovid algorithm was developed to advise people whether or not they were at particularly high risk from COVID-19, and people at high risk were asked to "shield" [12] Shielding advice during 2020 before the development of QCovid was based on individual health factors (non-gender specific) and in February 2021 (which was in the

middle of the survey fieldwork period) the QCovid algorithm was applied to GP data and shielding letters based on this algorithm were sent out. In 2021 the GP Patient Survey asked "At any time over the last 12 months, have you or someone you live with shielded at home due to being vulnerable to COVID-19 as a result of pre-existing health issues?", with response options "Yes, I have shielded", "Yes, someone else in my household has shielded", and "No". These responses will include people who were advised to shield both as a result of individual health conditions (pre-2021) and as a result of the implementation of the QCovid algorithm (February 2021 onwards).

*4.5. Demographic Characteristics*

The GP Patient survey also asks questions about age (with nine response options, "16–24", then in 10 year age groups until "85+"), ethnicity, and sexual orientation. Information for each respondent on GP Practice, Region and index of multiple deprivation (IMD), a small area level measure of deprivation are also available [24].

## 5. Language and Terminology

The analysis describes the health and healthcare experiences of people who are trans and/or non-binary, compared with people who are not. We note that the question asked in the survey did not include any specific wording asking if someone was trans and that included respondents may also include people with variation in sex characteristics, where their sex registered at birth and gender are different (more information in Box 1).

**Box 1.** Intersex survey respondents.

> Variations in sex characteristics (VSC), also referred to as differences in sex development (DSD), include a wide group of conditions. People living with VSC may or may not choose to identify as intersex, and the majority of people with VSC identify as male or female [11].
> We acknowledge that people who are intersex will be included among the survey respondents identified in this analysis as trans (people who respond that their gender identity is not the same as the sex they were registered with at birth). People who are intersex may have specific, or different, health and healthcare needs, or experiences of primary care, from other survey responders with gender different from the sex they were registered with at birth. These cannot be explored separately in this analysis, and this is a limitation. Future work is required to improve data collection for people who are intersex.

## 6. Methodological Considerations

*6.1. Sample Size*

Of the 850,206 survey responses to the GP Patient Survey received in 2021 (overall survey response rate 35.3%), 6333 survey responses included either "Non-binary" or "Prefer to self-describe" to the question asking about gender, or reported that their gender identity was different from their sex registered at birth, or both. Of these, after excluding missing or "prefer not to say" responses, 5212 responses were complete for both questions. People who are non-binary or prefer to self-describe their gender may or may not identify as trans (Table 2).

**Table 2.** Sample size: Trans and/or non-binary respondents.

| Gender | Trans Status | | | Total |
|---|---|---|---|---|
| | Gender Identity Different from Sex Registered at Birth | Gender Identity the Same as Sex Registered at Birth | Prefer Not to Say/Missing Response | |
| Female | 1708 | | | 1708 |
| Male | 1971 | | | 1971 |
| Non-binary | 456 | 490 | 274 | 1220 |
| Prefer to self-describe | 120 | 467 | 460 | 1047 |
| Prefer not to say/missing response | 387 | | | 387 |
| Total | 4642 | 957 | 734 | 6333 |

*6.2. Recoding Gender and Trans Status Responses*

For the main analysis, in order to have a large enough sample size, we will aggregate into a single group all responses in which gender is reported as "non-binary" or "prefer to self-describe" or in which the respondent reports that their gender identity is different from their sex registered at birth in a single group. We will compare this single group with people who report that their gender is Male or Female and that their gender identity is the same as their sex registered at birth. This approach to our main analysis was chosen following consultation with our PPI panel. However, it is important to acknowledge that, for example, experiences of people who are non-binary, and trans, may vary, and people who are non-binary and trans may have different experiences from people who are non-binary, and are not trans, and supplementary analyses will split these all groups from Table 2 more accurately where sample size allows. Specifically, for statistical models where we are using trans status as a predictor variable in we will use a likelihood ratio test to explore whether using this more nuanced categorisation of trans and/or non-binary adults improves the fit of the model over and above models where all trans and/or non-binary adults are included in a single group.

*6.3. Missing Data and Online Responses*

Since paper survey respondents could endorse more than one option for each question about gender, and where they did these were recoded as missing, in our first exploration of missing data we will examine the characteristics of people with a missing answer to the question about gender in order to explore whether multiple coding of responses is potentially an important concern. We will also describe the characteristics of online and paper respondents to explore whether response mode should be included in the analysis model.

*6.4. Analysis*

*Research question 1: What are the demographic characteristics, health conditions, and health-care experiences of trans and/or non-binary adults in England?*

We will describe the age, gender (including non-binary, and prefer to self-describe responses), trans status (i.e., whether or not a survey respondent reports that their gender identity is the same as their sex registered at birth), ethnicity, sexual orientation, region of residence and IMD of local-area of survey respondents. We will also describe how these compare between people who are trans or non-binary (i.e., people who report either that their gender identity is different from their sex registered at birth, or that their gender is non-binary, or that they prefer to self-describe their gender, or a combination of these), and those who are not. We will calculate weighted percentages in this descriptive analysis.

In the second analysis we will describe the weighted unadjusted prevalence of long-term health conditions among people who are trans or non-binary and then in adjusted analysis, using logistic regression for each long-term condition, to account for differences in the characteristics of the trans and/or non-binary population in England with all other survey responders, and whether these explain any differences in long-term disease prevalence seen. We will calculate odds ratios and 95% confidence intervals, and recycled predictions.

Informal lists of trans-friendly GPs [25], may mean that the registration of trans patients with a GP tends to cluster within a small number of practices (and this is consistent with comments from the PPI panel who added that they would be more likely to stick with a good GP if they found one). In order to incorporate this clustering analytically we will use mixed models, to the extent that the data support the assumptions needed for these models.

In the third analysis exploring experiences of primary care we will describe patient experience outcomes across all five domains which are publicly reported by NHS England ("Your local GP Services", "Making an appointment", "Your last appointment", "Your health", "Overall experience") and we will use the binary indicators (positive

experience—usually good or very good vs. not) used in the public reporting (18 items in total, Table 1) [26].

In supplementary analysis we will also explore "very good" and "very poor" responses, as earlier research in a group of gender diverse young people in England found higher proportions of both positive and negative experiences [27], consistent with the experiences of people living with multiple long-term health conditions [28], and discussions with our PPI panel. We will use logistic regression for the main analysis and will consider analyses with and without a random effect for GP practice to explore experiences of people who are trans and/or non-binary both nationally a within GP practice.

*Research question 2: Was there any difference in whether people who are trans and/or non-binary had been asked to shield or not compared with all other survey responders?*

In order to explore whether the percentage of trans and/or non-binary adults who were asked to shield was higher or lower than expected, in unadjusted analysis, we will describe the percentage of people who are trans and/or non-binary who were asked to shield, and compare this with all other survey respondents. In adjusted analysis, using logistic regression we will explore whether any differences can be explained by differences in socio-demographic characteristics or long-term health conditions, and differences between men and women, in order to try to characterise the performance of shielding guidance among people who are trans or non-binary.

*Research question 3: Does the relationship between being trans and/or non-binary, and self-reported long-term mental health problems, autism and autistic spectrum disorder and learning disability vary by age, gender, ethnicity, deprivation, sexual orientation or region.*

In this analysis we will use adjusted logistic regression (adjusting for age, ethnicity, deprivation, sexual orientation and region) and include interaction terms to explore whether there are any differences compared with all other survey respondents in the relationship between being trans or non-binary and reporting long-term mental health problems, autism and autistic spectrum disorder, or learning disability, by these other characteristics.

### 6.5. Pre-Registration

This analysis has been pre-registered on OSF.IO Saunders, Catherine L. 2022. The health and healthcare outcomes of trans and non-binary adults in England. OSF. April 19. osf.io/n7apf.

### 7. Discussion

England is beginning to collect more inclusive gender and trans status data, as well as sex data, and these data collections are important to understand how to improve healthcare services for people who are trans and/or non-binary. The analyses in this protocol are descriptive, and seek to characterise the health and healthcare experiences, the implementation of risk prediction algorithms and long-term mental health, autistic spectrum conditions and learning disabilities for people who are trans and/or non-binary in England. The work will provide up to date and population-based evidence on mental health outcomes and healthcare experiences which will build on evidence emerging internationally from cohorts of trans adult about the inter-relationship of these two dimensions [29,30].

Non-response is a challenge for all survey research. For the GP Patient Survey the impact of non-response has been explored previously, and the survey found to be valid for comparison of patient experience between GP Practices [22], although respondents who have attended a GP within the last year are over-represented [31]. Differences in response rates by age, ethnicity and deprivation will be accounted for through the use of adjusted analyses which incorporate these measures.

It is possible that the inclusion of a well worded question asking about both gender and trans status may have meant that people who are trans and/or non-binary are differentially more likely to respond to this survey than for other surveys where these questions are not asked so well. For the purposes of these analyses comparing responses from people

who are trans and/or non-binary with all other survey respondents, this is less of a methodological concern.

People who are trans or non-binary may be less likely to be registered with a GP overall, perhaps due to poor previous healthcare experiences. Although GP registration in England is nearly universal, people who are not registered with a GP are not included in the survey sampling frame and this is a further limitation. Rates of homelessness are higher among LGBT young people [32], and GP registrations are lower among people who are homeless [33].

It is possible to speculate about possible relationships, and pathways, between being trans and/or non-binary and health and healthcare outcomes. The use of private primary care services, particularly online services, (both for trans-specific healthcare, and generally due to issues with access to NHS GP services) came up in discussion with our PPI panel as well. This is an important (and fast developing) dimension, but not one that can be addressed with these data, and remains out of scope for this work.

## 8. Conclusions

The work will be disseminated through formal academic channels (conferences and blogs) but more importantly through ongoing work with the public involvement panel. We will additionally write a policy and a separate data focused research brief to accompany the work.

**Author Contributions:** Conceptualization, C.L.S., J.L., A.M.M., M.R., J.S. and R.D.; methodology, C.L.S., J.L., A.M.M., M.R., J.S. and R.D.; writing—original draft preparation, C.L.S.; writing—review and editing, C.L.S., J.L., A.M.M., M.R., J.S. and R.D.; funding acquisition, C.L.S. All authors have read and agreed to the published version of the manuscript.

**Funding:** C.L.S. holds a Career Development Fellowship (award number: MH041) funded as part of the Three NIHR Research Schools Mental Health Programme [*]. A.M.M. is funded by the EU/EFPIA Innovative Medicines Initiative Joint Undertaking BigData@Heart grant 116074. This research was supported by core funding from the: British Heart Foundation (RG/13/13/30194; RG/18/13/33946), and NIHR Cambridge Biomedical Research Centre (BRC-1215-20014) [*]. * The views expressed are those of the author(s) and not necessarily those of the NIHR or the Department of Health and Social Care.

**Institutional Review Board Statement:** Full details of the ethical approvals and guidance, including consent, and legal frameworks, under which the GP Patient Survey data are collected are given in detail on the study website https://www.gp-patient.co.uk/confidentiality (accessed on 28 June 2022). Since the primary purpose of the survey is service evaluation—to evaluate the service provided by GPs to their patients the survey does not require formal medical research ethical approval. GP Patient Survey data have been shared with the University of Cambridge under a Data Sharing Agreement with NHS England. According to guidance from the Health Research Authority, the secondary data analyses that we are planning in this work are exempt from review by a research ethics committee. The second important dimension is around PPI. Although public contributors/experts by experience are part of the research team, and not research subjects, and so ethical approval is not required, by working with the LGBTQ+ panel, we are asking people to disclose protected characteristics. When we first set up this panel we obtained ethical approval for the original work to acknowledge this tension but would not plan to do so for ongoing work now with this established group [8].

**Informed Consent Statement:** Not applicable.

**Data Availability Statement:** Data cannot be shared, although the analysis tool on the website www.gp-patient.co.uk (accessed on 28 June 2022) is very good, and interested researchers can use this to explore cross tabulations of the data.

**Conflicts of Interest:** The authors declare no conflict of interest.

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
