# Peer review of "The Health and Healthcare Outcomes of Trans and/or Non-Binary Adults in England: Protocol for an Analysis of Responses to the 2021 GP Patient Survey"

_sexes, doi:10.3390/sexes3030025_

Round 1

Reviewer 1 Report

The submission „The Health and Healthcare Outcomes of Trans and Non-binary 2 Adults in England: Protocol for an Analysis of Responses to the 3 2021 GP Patient Survey” is a pre-registration of a secondary analysis of the 2021 GP patient survey

The manuscript comes with several issues.

  1. LGBTQ+ should be fully explained.
  2. Why is it not LGBT*I*Q+?
  3. Possible trans issues related to the QCovid model should be explained.
  4. The research questions are not clearly explained
  5. RQ 1 seems to be 3 RQ in one: 1) demographic characteristics, 2) health conditions, 3) healthcare experiences.
  6. RQ2: needs a rationale and clearer explanation. QCovid must be mentioned for clarity.
  7. RQ3: needs a rationale and clearer explanation
  8. “We acknowledge this as a limitation to this work (as it may also be for research in 112 other areas as well), but it is not one that it is possible to address within the scope of the 113 available funding for this project without asking for unremunerated labour by the re-114 searcher.” Unclear: which researcher? Many researchers are involved in the study.
  9. Chapter 3: please report response rates.
  10. For readers it would be much better do completely report the measures (items and answer options) in chapter 4.
  11. “and supplementary analyses will split these all groups from Table 1 216 more accurately where sample size allows.” What exactly does that mean. Please add rationale for sufficiently large sub-sample sizes and take statistical power into consideration.
  12. “In our first exploration of missing data we will examine the characteristics of people 220 with a missing answer to the question about gender to explore whether multiple coding 221 of responses is potentially an important concern. We will also describe the characteristics 222 of online and paper respondents to explore whether response mode should be included 223 in the analysis model.” Unclear, please specify criteria for those decisions.
  13. “Informal lists of trans-friendly GPs [25], may mean that the registration of trans pa-242 tients with a GP tends to cluster within a small number of practices (and this is consistent 243 with comments from the PPI panel who added that they would be more likely to stick 244 with a good GP if they found one). We will describe the extent to which this occurs, and 245 whether it is explained by other patient or regional differences, using mixed models, to 246 the extent that the data support the assumptions needed for these models.” This analysis seems to be unrelated to the research questions. To properly include it, it must be part of the RQ.
  14. “In unadjusted analysis we will describe the percentage of people who are trans and 264 non-binary who were asked to shield, and compare this with all other survey respondents. 265 In adjusted analysis using logistic regression we will explore whether any differences can 266 be explained by differences in socio-demographic characteristics or long-term health con-267 ditions, and differences between men and women, in order to try to characterise the per-268 formance of shielding guidance among people who are trans or non-binary.” Rationale behind this analysis lacking.
  15. “In this analysis we will use adjusted logistic regression and include interaction terms 273 to explore whether there are any differences compared with all other survey respondents 274 in the relationship between being trans or non-binary and reporting long-term mental 275 health problems, autism and autistic spectrum disorder, or learning disability, by these 276 other characteristics” Unclear: adjustments for which variables?
  16. Discussion does not address practical implications
  17. Review board statement uses “I” language, even though an author group is mentioned
  18. Informed consent statement missing
  19. Data availability statement missing
  20. Conflicts of interest statement does not address conflicts of interest
  21. “Action for Trans Health. List of trans friendly GPs https://actionfortranshealth.org.uk/resources/for-trans-people/list-of-trans-375 friendly-gps/. 2021 „ List is not labeled correctly and hyperlink is not working
  22. To improve the quality of the preregistration it should refer to and follow other more detailed and successfully peer-reviewed preregistrations.

Author Response

Please see attached word document "Sexes reviewer responses"

Reviewer 2 Report

1) the paper switches being 'trans and non-binary' and 'trans and/or non-binary'. The latter would be preferable throughout given many non-binary people also describe themselves as trans

2) 'Gender and gender modality' might be preferable to 'gender identity and trans status'. See Florence Ashley's work on gender modality. 

3) the term 'gender reassignment' is outdated

4) the question 'is your gender identity the same as your sex' inherently presumes sex and gender are one and the same

Author Response

(The authors gave the same response as above.)

Reviewer 3 Report

Thank you for the opportunity to review this study protocol on trans and non-binary people’s health and healthcare outcomes based on the data from 2021 GP Patient Survey. There are several strengths of this paper, such that it is well-written (e.g., concise while capturing sufficient detail), the description of table is clear, and the discussion section provides helpful information on proposed analyses to be undertaken. I understand that the authors are at an early stage of analysing the data so this paper may not be the best place to address some of my comments on data presentation – if that is the case, feel free to defer these to the authors’ future work. Below I list more specific points that the authors may consider addressing, along with references to material that might be useful to the authors in responding to these concerns.

Broad general comment: I was confused by the iteration of  “LGBTQ+” throughout the paper for a study that focuses on trans and non-binary people’s experiences. There is a history of tokenistic inclusion of trans and non-binary people under this broader umbrella term in the academic literature. Have the authors thought about only referencing trans and non-binary people in this paper?

1.      Line 26: The sentence for the third research question should end with a question mark (?) Same for Line 88.

2.      Line 32: The framing of “poor health” and “poor healthcare experiences” for trans and non-binary people should be contextualised within the disparity context compared to cisgender people. The authors can consider citing recent studies of other parts of the world (see Kattari et al. 2022; Treharne et al., 2022) to demonstrate international relevance and the need for such study in England. A sentence or two about what the present study adds to the nationwide 2012 UK Trans Mental Health Study (Ellis et al., 2015) may also help with building up the local context.

Kattari SK, Bakko M, Hecht HK, Kattari L. Correlations between healthcare provider interactions and mental health among transgender and nonbinary adults. SSM Popul Health. 2020;10:100525

Treharne GJ, Carroll R, Tan KKH, Veale JF, Supportive interactions with primary care doctors are associated with better mental health among transgender people: results of a nationwide survey in Aotearoa/New Zealand, Family Practice, 2022 https://doi.org/10.1093/fampra/cmac005

3.      Something is missing for the sentence from Line 36-39. Disadvantaged “identities”?

4.      Line 143 – In the England/broader UK? Be careful about overclaiming.

5.      Line 145 – Tested with “trans and non-binary” patients specifically?

6.      Line 157 (and section 6.3) – Treating paper survey respondents who selected more than one gender options as missing doesn’t sound ethical. While I understand that some may be troll responses, there is indeed a chance that non-binary folks may select more than one options because they thought all options are applicable to them. There is not an absolute right way of carrying out the gender analysis, but I encourage the authors to be transparent with their decision-making process.

7.      Table 1: Is there a reason why the data for “gender identity the same as sex registered at birth” not shown for female and male? The inclusion of percentage can provide a clearer depiction of the different gender representation.

8.      Line 208: Were there considerations for classifying some responses under “prefer to self-describe” as female, male, trans female, or trans male?

9.      Line 238. The authors can consider including the unit of analysis next to logistic regression (odds ratio?)

10.    Line 220: “sociodemographic” characteristics specifically? There is no need to cite this particular article (Cunningham & Wells, 2017) but the authors can read about how age is a predictor for non-responses from this article:

Cunningham M, Wells M. Qualitative analysis of 6961 free-text comments from the first National Cancer Patient Experience Survey in Scotland. BMJ Open 2017;7:e015726.

11.    The discussion session can end with a more positive note about how the findings can be used to inform policy works in improving the health and healthcare access of trans and non-binary people in English (and broader UK). What are some of the planned dissemination methods of findings?

Author Response

(The authors gave the same response as above.)

Round 2

Reviewer 1 Report

The action letter and revision have sufficiently addressed my concerns.